# Duodenal Ulceration in a Child with Coeliac Disease

**DOI:** 10.3390/diagnostics10010031

**Published:** 2020-01-09

**Authors:** Polina S. Lototskaya, Marina A. Manina, Aleksandr S. Tertychnyy, Andrey A. Zamyatnin, Svetlana I. Erdes

**Affiliations:** 1Filatov Clinical Institute of Child Health, Sechenov First Moscow State Medical University, 119991 Moscow, Russia; lotops@bk.ru (P.S.L.); manina12345@mail.ru (M.A.M.); 2Department of Pathological Anatomy, Sechenov First Moscow State Medical University, 119991 Moscow, Russia; atertychnyy@yandex.ru; 3Institute of Molecular Medicine, Sechenov First Moscow State Medical University, 119991 Moscow, Russia; 4Belozersky Institute of Physico-Chemical Biology, Lomonosov Moscow State University, 119992 Moscow, Russia

**Keywords:** celiac disease, *Helicobacter pylori*, peptic ulcer disease

## Abstract

Coeliac disease (CD) is a gluten-dependent inflammatory disease of the small bowel that affects up to 1% of the global population. Herein, the presence of ulcers, erosions, or strictures in the duodenum for non-advanced cases of CD is a rarity. Case report: We present a clinical case of a 17-year-old girl, who from the age of 9, had suffered from erosive *Helicobacter pylori* (HP)-associated gastritis and erosive duodenitis. At 16, she was diagnosed with a duodenal ulcer, complicated by cicatricial deformity of the bulb. While an atypical course in the development of the disease had led to the initial delay in diagnosis, a serum study and an intestinal biopsy confirmed CD. Discussion: A recent study found an elevated rate of peptic ulcer disease in patients with CD. From literature searches, comorbid HP infection and CD have indeed been widely reported, whereas cases highlighting the prevalence of CD-associated peptic ulcers have been observed and reported in only a few instances. Consequently, greater awareness is warranted and must be exercised for identifying the origins of ulcerative lesions that may be CD-related or -derived.

## 1. Introduction

Coeliac disease (CD) is an immunity-mediated systemic disease caused by the intake of gluten and related prolamins from cereals of wheat, barley, and rye in genetically susceptible people [1,2]. The diagnosis of CD in children remains a clinical problem because of the variety of ways in which CD can symptomatically manifest itself. Thus, it is important to investigate and diagnose CD not only in children with obvious gastrointestinal symptoms but also in children with less clear clinical manifestations because of the potential risk it poses to general health at a later stage in life [3,4,5,6]. In older children and adolescents, CD can be identified by numerous non-gastrointestinal symptoms, named as ‘‘atypical symptoms’’, which are in strong contrast to ‘‘typical symptoms’’ that have been previously described as being gastrointestinal and extraintestinal in origin [7,8]. However, these terms are now considered obsolete and are no longer recommended for such a use [9]. Following on, it has become more important to detect CD in “at-risk” groups comprising persons with genetic predispositions or through the analysis of medical history, for individuals with iron deficiency anemia, premature osteoporosis or osteopenia, type 1 diabetes mellitus, autoimmune thyroid disease, liver disease, and Down syndrome or Turner’s syndrome [10].

A large number of studies have reported a significant increase in the frequency of the atypical form of CD, ranging between 25% and 93% [11] of patients with non-specific symptoms and resulting in undiagnosed patients (suffering from CD) having to visit a number of multiple medical specialists. Consequently, the ratio of diagnosed to unidentified cases of CD in some countries has reached as high as 1:10 [12]. Whereas a lot of patients had been categorized to have reduced symptoms or no symptoms at all, many were positively diagnosed as a result of investigating the prevalence of atypical symptoms [13]. Typically, this is because duodenal ulcers are not a common feature of CD and the prevalence of these, with or without erosions or strictures within the duodenum for non-advanced cases of CD, are a rarity. There is a scarceness of reported data relating peptic disease (PD) to CD and, consequently, further investigations are indeed warranted to offer greater clarity in diagnosing CD more effectively.

## 2. Case Presentation

Herein, we report a case of CD defined through a combined number of approaches as atypically symptomatic of CD and which, prior to this, had resulted in it on a long-term basis. The case revolves around a 17-year-old girl, who from the age of 9, had suffered from erosive *Helicobacter pylori* (HP)-associated gastritis and erosive duodenitis confirmed endoscopically, histologically, and by a rapid urease biopsy test. During this period, the indicators of weight and height were age-appropriate and sometimes the patient suffered headaches, an unstable mood, physical and mental weakness, and fatigue. A triple course of bismuth-based eradication therapy with amoxicillin and clarithromycin was recommended throughout this period.

At the age of 16, the girl was diagnosed with a duodenal ulcer, complicated by cicatricial deformity of the bulb and chronic HP-positive gastritis. Serum examinations revealed the positive presence of the anti-transglutaminase IgA antibody (tTG) 415 U/mL (reference values <20 U/mL), positive anti-EMA IgA, positive anti-EMA IgG and anti-DGP IgA 183 U/mL (reference values <10 U/mL), anti-DGP IgG 131 U/mL (reference values <10 U/mL) and severe subtotal villous atrophy, as suggested by duodenal biopsy (stage 3B of the Marsh–Oberhuber classification; Figure 1A). The results of this survey confirmed the diagnosis of CD, according to the ESPGHAN Guidelines for Diagnosis of Coeliac Disease [3].

Therapy included adherence to a gluten-free diet and quadruple eradication therapy with bismuth, a proton pump inhibitor, doxycycline, and metronidazole. Laboratory examinations and tests after 3 months revealed anti-transglutaminase IgA antibody (tTG) levels of 173 U/mL (reference values <20 U/mL), positive anti-EMA IgA, negative anti-EMA IgG, anti-DGP IgA levels of 14.8 U/mL (reference values <10 U/mL), anti-DGP IgG levels of 14.9 U/mL (reference values <10 U/mL), and normal villous architecture with crypt hyperplasia returning to normal based upon histological examination (stage 2 of the Marsh–Oberhuber classification; Figure 1B). ELISA test for the detection of *H. pylori* antigen in stool showed a negative result. Clinical improvement was observed, including the restoration of a good state of mental and physical health. An endoscopic analysis revealed the scarring of the duodenal ulcer, suggesting its remission.

The study was performed according to the Helsinki Declaration of 1975, as revised in 2008 and approved by the Ethics Committee at Sechenov University (N04-04 from 5 April 2017) prior to this study. A written informed consent was obtained from the parents of patient.

## 3. Discussion

Duodenal ulcers were not believed to be a characteristic feature of CD, which until recently was considered to be a rare condition. However, Dickey and Hughes (2004), while undertaking upper endoscopy observations on 1200 CD patients during a 2-year period (from April 2001 through March 2003), revealed erosions in the second part of the duodenum for five patients that were otherwise normal for any pathological changes of the stomach or the duodenal bulb [14]. Contrastingly, Molina-Infante et al. (2010) presented a 43 year-old female patient, with patchy erosions within the duodenal bulb [15] and who had a long-standing history of rheumatoid arthritis, iron deficiency anemia, and was being treated with methotrexate. Generally speaking, duodenal ulcers were commonly recorded after a long asymptomatic period following the diagnosis of CD and were considered a complication of CD, particularly as ulcers, and erosions of the small bowel had been commonly described in advanced cases of CD and mainly in adults. 

Subsequent efforts related to detecting CD in the pediatric population have looked at the development of ulcers through identifying acute abdominal pain and whether this coincides with the detection of CD. Consequently, Mones and Mercer (2011) and Galloway et al. (2013) presented clinical cases of such adolescents, and Veres et al. (2011) reported a case of a 6-year-old girl who had a sudden onset of hematemesis caused by a duodenal ulcer [16,17,18]. Collectively, these reports had been documented with the prevalence of duodenic ulcers at the time of diagnosis and all patients were HP-negative.

In 2009, the authors of a study retrospectively analyzed and reported the results of all the endoscopies performed on children and young adults diagnosed with CD between the years 2004 and 2008, within the medical centers of Tel-Aviv and Haifa [19]. From a cohort of 240 confirmed CD patients, duodenal PD was documented in 22 patients (9.2%), 16 (73%) of which were HP-negative and most of whom were below the age of 20 years. Of these 22 patients, 15 had erosions and 7 had duodenal ulcers within the duodenal bulb. The authors concluded that PD was not uncommon when being presented with cases of CD and that the onset of it may even occur during infancy. They suggested including CD in the differential diagnosis of patients with non-HP PD and suggested routine CD serology and small bowel biopsies from patients with unexplained PD.

Much later, an article by Tumgor et al. (2018) reported the results from the endoscopic examination of 1769 children showing a variety of gastrointestinal symptoms recorded between the years 2012 and 2017 [20]. Herein, CD was diagnosed positively through serology, endoscopy, and histopathology tests in 250 (14%) of all patient cases and peptic ulcers diagnosed in 74. CD-positive serology was determined in 22 of the 74 (29%) and of which 63% were HP-negative. Collectively, such findings suggest a strong correlation between the prevalence of peptic ulcers with CD and approximately half of which were HP-negative cases.

In conclusion, our reported case of ulcerative lesions of the duodenum (without any other manifestations of CD for eight years), may present itself as a peculiarity and is likely to be a consequence of a late diagnosis rather than due to the delayed onset of CD. In support, we highlight recent studies that have reported elevated frequencies of peptic ulcer disease in patients with CD. Consequently, a possible lack of awareness of the different clinical manifestations of CD may contribute to prolonged delays in diagnosis and treatment. Therefore, a greater level of awareness relating to the atypical endoscopic features of CD is indeed warranted. This may have greater importance if other (more frequent) causes of peptic ulcers, such as HP infection or non-steroid anti-inflammatory drugs have already been precluded. Moreover, (and on a precautionary note,) it may even be recommended that CD should be investigated by default, even if patients with peptic ulcers are HP-positive. 

## Figures and Tables

**Figure 1 diagnostics-10-00031-f001:**
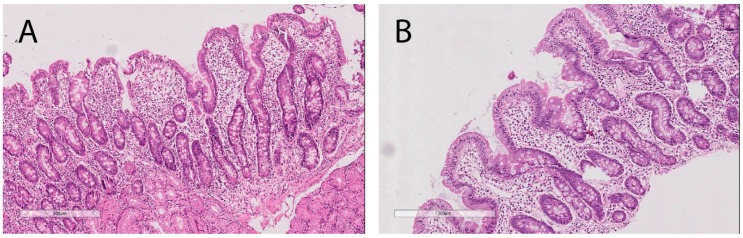
Histological examination of the duodenal biopsy. (**A**) Before gluten-free diet. Marsh stage 3b lesion of gluten-induced enteropathy characterized by broad, blunted villi, crypt elongation and increase in intra-epithelial lymphocytes (IELs). (**B**) Morphological improvement under gluten-free diet due to a return to normal villous architecture and a decrease in intra-epithelial lymphocytes (IELs). H&E. ×100.

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
