# Peer review of "Duodenal Ulceration in a Child with Coeliac Disease"

_diagnostics, 2020, doi:10.3390/diagnostics10010031_

Round 1

Reviewer 1 Report

Comments:

This manuscript describes the clinical case of a patient with celiac disease, who additionally suffered from erosive Helicobacter pylori (HP)-associated gastritis and erosive duodenitis. At the age of 16 she was diagnosed with a duodenal ulcer with complications and CD. The authors lay out a well-organized study but I have some comments/suggestion that need to be explained before the manuscript will be published:

- introduction: is quite lapidary, even scanty… I understand that this is a Case report, but for non-specialist in the field of gastroenterology some issues may be unclear. I suggest the Authors to describe a short explanation what the typical CD symptoms are, and then also provide some examples of atypical symptoms. Please also provide relevant literature

- line 41: PD please explain this abbreviation when it appear for the first time in the text

- line 42: I suggest to entitled this paragraph “Case Presentation”

- line 46-48: presenting research in form of case report needs as much detail as possible, therefore I suggest to provide more details. Based on which tests/methods the erosive Helicobacter pylori (HP)-associated gastritis and erosive duodenitis was determined? Also it is unclear when eradication therapy was it carried out - at the age of 9 or 17? what kind of treatment was used?

- line 47-48: “…the patient suffered headaches, unstable mood, physical and mental weakness and fatigue” – such symptoms/manifestations could indicate CD.  Has the patient's mental and physical state of health improved after 3 months of therapy with GFD?

- line 50-54: laboratory examinations revealed the positive presence of serological markers of CD  and severe subtotal villous atrophy - please indicate precisely that the results of these tests allowed to finally diagnose the CD, and quote the suitable references with international algorithm for CD diagnosis

- line 55-60: As indicated by the results of the study, the GFD therapy effectively reduced the anti-tTG, -EMA and -DGP antibody as well as enabled reconstruction of the intestinal mucosa functioning, but what about other health problems? How this therapy affected duodenal ulcers ect.? Please provide more information

-line 106: ”This may have great importance if other (more frequent) causes of CD, such as H. pylori infection or NSAIDs” – unclear, this statement suggest that H. pylori infection or NSAIDs  are the triggers of CD – rewrite please

Author Response

This manuscript describes the clinical case of a patient with celiac disease, who additionally suffered from erosive Helicobacter pylori (HP)-associated gastritis and erosive duodenitis. At the age of 16 she was diagnosed with a duodenal ulcer with complications and CD. The authors lay out a well-organized study but I have some comments/suggestion that need to be explained before the manuscript will be published:

Our response:

We are thankful to the reviewer for the positive comments relating to our manuscript. Here follow our point-by-point replies to the raised queries.

- introduction: is quite lapidary, even scanty… I understand that this is a Case report, but for non-specialist in the field of gastroenterology some issues may be unclear. I suggest the Authors to describe a short explanation what the typical CD symptoms are, and then also provide some examples of atypical symptoms. Please also provide relevant literature

Our response:

The Introduction was expanded as suggested by the reviewer (please see the lines 32-56).  

- line 41: PD please explain this abbreviation when it appear for the first time in the text

Our response:

Done  (please see the line 54)

- line 42: I suggest to entitled this paragraph “Case Presentation”

Our response:

Done (please see the line 57)

- line 46-48: presenting research in form of case report needs as much detail as possible, therefore I suggest to provide more details. Based on which tests/methods the erosive Helicobacter pylori (HP)-associated gastritis and erosive duodenitis was determined?

Our response:

The information requested was added (please see the lines 60-62)

Also it is unclear when eradication therapy was it carried out - at the age of 9 or 17? what kind of treatment was used? Bismuth, proton pump inhibitor, doxycycline, Metronidazole

Our response:

Requested information was added (please see lines 64-65 and the lines 74-75)

- line 47-48: “…the patient suffered headaches, unstable mood, physical and mental weakness and fatigue” – such symptoms/manifestations could indicate CD.  Has the patient's mental and physical state of health improved after 3 months of therapy with GFD?

Our response:

Requested information was added (please see lines 81-83)

- line 50-54: laboratory examinations revealed the positive presence of serological markers of CD  and severe subtotal villous atrophy - please indicate precisely that the results of these tests allowed to finally diagnose the CD, and quote the suitable references with international algorithm for CD diagnosis

Our response:

Requested information was added (please see the lines 71-73)

- line 55-60: As indicated by the results of the study, the GFD therapy effectively reduced the anti-tTG, -EMA and -DGP antibody as well as enabled reconstruction of the intestinal mucosa functioning, but what about other health problems? How this therapy affected duodenal ulcers ect.? Please provide more information

Our response:

Requested information was added (please see the line 80)

-line 106: ”This may have great importance if other (more frequent) causes of CD, such as H. pylori infection or NSAIDs” – unclear, this statement suggest that H. pylori infection or NSAIDs  are the triggers of CD – rewrite please

Our response:

The sentence was rewritten (please see the lines 131-135)

Reviewer 2 Report

This manuscript describes the clinical case of a patient with celiac disease, who additionally suffered from erosive Helicobacter pylori (HP)-associated gastritis and erosive duodenitis. I have some comments/suggestion that need to be explained before the manuscript will be published

The introduction needs to be strengthened. More reviews should be described.

What wrer the positive presence of serological markers of CD and severe subtotal villous atrophy

When was the H.pylori eradicated?

English needs to be re-edited

Author Response

This manuscript describes the clinical case of a patient with celiac disease, who additionally suffered from erosive Helicobacter pylori (HP)-associated gastritis and erosive duodenitis. I have some comments/suggestion that need to be explained before the manuscript will be published

We are thankful to the reviewer for the positive comments relating to our manuscript. Here follow our point-by-point replies to the raised queries.

The introduction needs to be strengthened. More reviews should be described.

Our response:

The Introduction was expanded as suggested by the reviewer (please see lines 32-56).  

What wrer the positive presence of serological markers of CD and severe subtotal villous atrophy

Our response:

The information requested was added (please see the lines 67-71)

When was the H.pylori eradicated?

The information requested was added (please see the lines 80-81)

English needs to be re-edited

We are very grateful to Dr. Surinder M. Soond for his additional editorial work throughout the preparation of the revised version of the manuscript.

Round 2

Reviewer 1 Report

The manuscript has been corrected and all my comments were taken into consideration. I think it is ready for publication in this version

Reviewer 2 Report

I have no further questions. The revision is good enough!